# Synthesis and Biochemical Evaluation of 8*H*-Indeno[1,2-*d*]thiazole Derivatives as Novel SARS-CoV-2 3CL Protease Inhibitors

**DOI:** 10.3390/molecules27103359

**Published:** 2022-05-23

**Authors:** Jing Wu, Bo Feng, Li-Xin Gao, Chun Zhang, Jia Li, Da-Jun Xiang, Yi Zang, Wen-Long Wang

**Affiliations:** 1School of Life Sciences and Health Engineering, Jiangnan University, Wuxi 214122, China; 6191504014@stu.jiangnan.edu.cn (J.W.); lxgao@simm.ac.cn (L.-X.G.); zhangchun@jiangnan.edu.cn (C.Z.); 2School of Life Science and Biopharmaceutics, Shenyang Pharmaceutical University, Shenyang 110016, China; fengboo@live.com (B.F.); jli@simm.ac.cn (J.L.); 3State Key Laboratory of Drug Research, Shanghai Institute of Materia Medica, Chinese Academy of Sciences, Shanghai 201203, China; 4Zhongshan Institute for Drug Discovery, Shanghai Institute of Materia Medica, Chinese Academy of Sciences, Zhongshan Tsuihang New District, Zhongshan 528400, China; 5Xishan People’s Hospital of Wuxi City, Wuxi 214105, China

**Keywords:** COVID-19, M^pro^ inhibitors, drug design and synthesis, structure-activity relationships (SAR)

## Abstract

The COVID-19 pandemic caused by SARS-CoV-2 is a global burden on human health and economy. The 3-Chymotrypsin-like cysteine protease (3CL^pro^) becomes an attractive target for SARS-CoV-2 due to its important role in viral replication. We synthesized a series of 8*H*-indeno[1,2-*d*]thiazole derivatives and evaluated their biochemical activities against SARS-CoV-2 3CL^pro^. Among them, the representative compound **7a** displayed inhibitory activity with an IC_50_ of 1.28 ± 0.17 μM against SARS-CoV-2 3CL^pro^. Molecular docking of **7a** against 3CL^pro^ was performed and the binding mode was rationalized. These preliminary results provide a unique prototype for the development of novel inhibitors against SARS-CoV-2 3CL^pro^.

## 1. Introduction

The global pandemic of coronavirus disease (COVID-19) caused by the severe acute respiratory syndrome coronavirus 2 (SARS-CoV-2) has posted major challenges to public health systems and the economy worldwide [1,2,3,4,5]. There have been 434 million confirmed cases of COVID-19 worldwide as of the end of February 2022, and almost 6 million deaths have been reported [6]. Although multiple effective vaccines against COVID-19 are available, reinfections and breakthrough infections are frequently reported [7,8]. In addition, the virus is continuing to evolve, and a new variant named Omicron enables the virus to evade the immune protective barrier due to a large number of mutations in the receptor binding sites [9,10,11]. Therefore, it is urgent to develop effective drugs and specific treatments for people who are infected by COVID-19 with severe symptoms.

3CL^pro^ (also called M^pro^) plays an essential role during replication and transcription of SARS-CoV-2 and has been regarded as an attractive target for treating COVID-19 and other coronavirus-caused diseases [12,13,14]. The development of 3CL^pro^ inhibitors has attracted much attention from medicinal chemists and the pharmaceutical industry. The collective efforts culminated in the recent approval of Paxlovid (nirmatrelvir) by FDA for the treatment of SARS-CoV-2 [15]. As shown in Figure 1, Most known 3CL^pro^ inhibitors are peptidomimetic inhibitors containing a warhead of Michael acceptor, such as nirmatrelvir with nitrile [16], YH-53 with benzothiazolyl ketone [17], compound **1** with α-ketoamide [18], and compound **2** with aldehyde [19]. Others are nonpeptidic inhibitors including covalent and noncovalent inhibitors. Covalent inhibitors, such as Carmofur, Shikonin [20], and **3** [21], are identified by high-throughput screening. Noncovalent inhibitor CCF0058981 [22] and flavonoid analogs (baicalin, baicalein, and 4′-*O*-Methylscutellarein) [23,24] were obtained through structure-based optimization and from traditional Chinese medicines, respectively.

In pursuit of novel 3CL^pro^ inhibitors, we identified 8*H*-indeno[1,2-*d*]thiazole derivative **4** as a novel SARS-CoV-2 3CL^pro^ inhibitor (IC_50_ = 6.42 ± 0.90 μM) through high-throughput screening of our compound collection (Figure 2). This result provided us with an opportunity to explore novel small molecule inhibitors against SARS-CoV-2 3CL^pro^. Herein, we designed and synthesized a series of 8*H*-indeno[1,2-*d*]thiazole derivatives, evaluated their inhibitory activities against SARS-CoV-2 3CL^pro^, and elucidated the SARs. Selected compound **7a** was subjected to molecular docking to predict the binding mode with SARS-CoV-2 3CL^pro^.

## 2. Results and Discussion

### 2.1. Design and Synthesis of 8H-Indeno[1,2-d]thiazole Derivatives

Based on the structure of compound **4**, 14 new 8*H*-indeno[1,2-*d*]thiazole derivatives (compounds **7a**–**7l**, and **10a**–**10b**) (shown in Figure 1 and Figure 2) were designed and synthesized through a two-step synthesis from the appropriate ketone and thiourea [25,26,27,28]. Adjusting the methoxy group of compound **4** from position 5 to position 6 afforded compound **7a**. Considering the effects of steric hindrance and electron withdrawing, compounds **7b**–**7e** were synthesized by substitution of the methoxy group for the butoxy, isobutoxy, and methyl groups and for the chlorine atom. After replacing the 3,5-dimethoxybenzamido moiety in compound **7a** with 3,4,5-trimethoxybenzamido, 3,5-diacetoxybenzamido, 3-methoxybenzamido, 3-fluorobenzamido, thiophene-2-carboxamido, and 4-chlorobenzamido, compounds **7f**–**7k** were obtained. To evaluate the effect of ring expansion, compound **7l** was synthesized. Finally, ring opening analogues **10a** and **10b** were synthesized to elucidate the effect of the central ring on the inhibition of 3CL^pro^.

### 2.2. SARS-CoV-2 3CL^pro^ Inhibitory Activities and Structure-Activity Relationships

All synthesized compounds were evaluated for inhibitory activity against SARS-CoV-2 3CL^pro^ using PF-07321332 as positive control [29,30,31], and the results were detailed in Table 1. We initially prepared **7a** from the commercially available compound **5a** by the route outlined in Figure 1. We noticed that compound **7a** with 6-methoxy group on the phenyl ring exhibited inhibitory activity against SARS-CoV-2 3CL^pro^ with 1.28 ± 0.17 μM, about five times more potent than compound **4** with 5-methoxy group on the phenyl ring. The result indicated that the position of the methoxy group on the phenyl ring significantly affected inhibitory activities against SARS-CoV-2 3CL^pro^. To explore the SAR of this seemingly important position, methoxy group on compound **7a** was replaced by butoxy (**7b**), isobutoxy (**7c**), methyl groups (**7d**), and chlorine atom (**7e**); the inhibitory activities of the corresponding compounds **7b**–**7e** were completely abolished. These results demonstrated that the effect of steric hindrance at this position was detrimental to inhibitory activities. The SAR of **R^3^** was explored next. Replacement of the 3,5-dimethoxybenzamido moiety with 3,4,5-trimethoxybenzamido moiety, 3,5-diacetoxybenzamido moiety, 3-methoxybenzamido moiety, 3-fluorobenzamido moiety, thiophene-2-carboxamido moiety, and 4-chlorobenzamido moiety led to compounds **7f**, **7g**, **7h**, **7i**, **7j**, and **7k**, respectively. The inhibitory activity of compounds **7f** and **7g** dropped significantly, while compound **7h** almost maintained its inhibitory activities. These results indicated that the extra steric hindrance had negative impact on the inhibitory activities. Compared to compound **7h**, the inhibitory activities of compounds **7i**–**7k** diminished; these results indicated that introduction of an electron-withdrawing group or heterocyclic ring on the scaffold of 8*H*-indeno[1,2-*d*]thiazole took negative roles for inhibitory activities. Expanding the five-membered ring on compound **7a** to a six-membered ring led to compound **7l**, which unfortunately did not show any inhibitory activity against SARS-CoV-2 3CL^pro^. Opening the five-membered ring on compound **7a** resulted in compounds **10a** and **10b**, which also lost inhibitory activities. These results indicated that the five-membered ring on compound **7a** is important for the inhibitory activity against SARS-CoV-2 3CL^pro^.

### 2.3. Predicting Binding Mode of ***7a*** with 3CL^pro^

To explore the interaction mode between small molecule **7a** and 3CL^pro^ (PDB code: 6M2N) [23], we carried out molecular docking by applying AutoDock 4.2 program [31,32,33,34]. Figure 3a showed that **7a** docked well into the binding pockets S1 and S2 of 3CL^pro^, in which the S1, S2 sites play a key role in substrate recognition [35]. As illustrated in Figure 3b, the indene moiety of compound **7a** buried deeply into the hydrophobic S2 subsite with π-electrons with Arg188 and hydrophobic interaction with Met165; the 3,5-dimethoxybenzamido moiety of compound **7a** formed strong H-bonds with Asn142, Glu166 on S1 subsite, while compounds **4** and **7h** escaped from S1 subsite, as shown in Appendix A.

## 3. Materials and Methods

### 3.1. Chemistry

All chemical reagents are reagent grade and used as purchased. ^1^H NMR (400 MHz) spectra were recorded on a Bruker AVIII 400 MHz spectrometer (Bruker, Billerica, MA, USA). The chemical shifts were reported in parts per million (ppm) using the 2.50 signal of DMSO (^1^H NMR) and the 39.52 signal of DMSO (^13^C NMR) as internal standards. ESI Mass spectra (MS) were obtained on a SHIMADZU 2020 Liquid Chromatograph Mass Spectrometer (SHIMADZU, Kyoto, Japan).

#### 3.1.1. General Procedure for the Synthesis of Compounds **7a**–**7k** (Exemplified by **7a**)

To a solution of **5a** (6.2 mmol, 1.0 equiv) in dry ethanol (25 mL) were added thiourea (12.4 mmol, 2.0 equiv) and bromine (6.8 mmol, 1.1 equiv) at room temperature. The reaction solution was stirred at 100 °C for 5–6 h, At the end of the reaction, the solvent was evaporated, and aqueous ammonium hydroxide (25%) was added to the residue. The precipitated solid was collected without purification for the next step. The mixture of **6a** (2.2 mmol, 1.1 equiv), aromatic acid (2.0 mmol, 1.0 equiv), HATU (2.0 mmol, 1.0 equiv), and DIPEA (6.0 mmol, 3.0 equiv) in DMF (15 mL) was stirred at room temperature for 2 h. The reaction mixture was quenched with water. The aqueous layer was extracted with EtOAc (30 mL × 2). The combined organic layers were dried over Na_2_SO_4_. The residue was purified by column chromatography on silica gel (eluting with DCM) to afford compound **7a** as a yellow solid (280.0 mg, yield 37%). ^1^H NMR (400 MHz, DMSO-*d*_6_) δ 12.81 (s, 1H), 7.46 (d, *J* = 8.4 Hz, 1H), 7.33 (d, *J* = 2.0 Hz, 2H), 7.22 (d, *J* = 2.0 Hz, 1H), 6.94 (dd, *J* = 8.0, 2.4 Hz, 1H), 6.74 (t, *J* = 2.0 Hz, 1H), 3.87 (s, 2H), 3.84 (s, 6H), 3.80 (s, 3H) ppm. ^13^C NMR (100 MHz, DMSO-*d*_6_) δ 164.22, 162.12, 160.52, 157.76, 155.03, 147.98, 133.82, 130.05, 128.39, 118.28, 112.37, 111.83, 105.74, 105.08, 55.60, 55.36, 32.43 ppm. MS (ESI): *m*/*z* calcd for C_20_H_19_N_2_O_4_S [M + H]^+^ 383.11, found 383.20.

*N-(6-butoxy-8H-indeno[1,2-d]thiazol-2-yl)-3,5-dimethoxybenzamide* (**7b**), eluting with DCM, yield = 32%; ^1^H NMR (400 MHz, DMSO-*d*_6_) δ 12.81 (s, 1H), 7.43 (d, *J* = 8.0 Hz, 1H), 7.33 (d, *J* = 2.4 Hz, 2H), 7.18 (s, 1H), 6.91 (dd, *J* = 8.4, 2.4 Hz, 1H), 6.73 (d, *J* = 2.4 Hz, 1H), 3.99 (t, *J* = 6.4 Hz, 2H), 3.84 (s, 2H), 3.83 (s, 6H), 1.74–1.67 (m, 2H), 1.49–1.40 (m, 2H), 0.95–0.91 (m, 3H) ppm. ^13^C NMR (100 MHz, DMSO-*d*_6_) δ 164.25, 162.15, 160.54, 157.20, 155.04, 147.96, 133.85, 129.95, 128.32, 118.29, 112.97, 112.39, 105.76, 105.10, 67.48, 55.62, 32.43, 30.89, 18.84, 13.76 ppm. MS (ESI): *m*/*z* calcd for C_23_H_25_N_2_O_4_S [M + H]^+^ 425.15, found 425.10.

*N-(6-isobutoxy-8H-indeno[1,2-d]thiazol-2-yl)-3,5-dimethoxybenzamide* (**7c**), eluting with DCM, yield = 40%; ^1^H NMR (400 MHz, DMSO-*d*_6_) δ 8.27 (s, 1H), 7.43 (dd, *J* = 8.4, 3.2 Hz, 1H), 7.36–7.32 (m, 2H), 7.19 (d, *J* = 2.8 Hz, 1H), 6.92 (dd, *J* = 8.4, 3.2 Hz, 1H), 6.73 (t, *J* = 2.4 Hz, 1H), 3.85 (s, 2H), 3.83 (s, 6H), 3.78 (d, *J* = 6.4 Hz, 2H), 2.06–2.00 (m, 1H), 0.99 (d, *J* = 7.2 Hz, 6H) ppm. ^13^C NMR (100 MHz, DMSO-*d*_6_) δ 164.25, 162.17, 160.50, 157.26, 154.99, 147.94, 133.86, 129.95, 128.31, 118.26, 112.99, 112.45, 105.74, 105.06, 74.11, 55.59, 32.41, 27.78, 19.11 ppm. MS (ESI): *m*/*z* calcd for C_23_H_25_N_2_O_4_S [M + H]^+^ 425.15, found 425.20.

*3,5-dimethoxy-N-(6-methyl-8H-indeno[1,2-d]thiazol-2-yl)benzamide* (**7d**), eluting with DCM, yield = 47%; ^1^H NMR (400 MHz, DMSO-*d*_6_) δ 12.81 (s, 1H), 7.45 (d, *J* = 7.6 Hz, 1H), 7.39 (s, 1H), 7.34 (d, *J* = 2.4 Hz, 2H), 7.18 (d, *J* = 7.6 Hz, 1H), 6.74 (t, *J* = 2.0 Hz, 1H), 3.87 (s, 2H), 3.84 (s, 6H), 2.38 (s, 3H) ppm; ^13^C NMR (100 MHz, DMSO-*d*_6_) δ 164.26, 162.14, 160.50, 155.22, 146.31, 134.42, 134.30, 133.78, 129.84, 127.40, 126.00, 117.56, 105.73, 105.09, 55.58, 32.16, 21.13 ppm. MS (ESI): *m*/*z* calcd for C_20_H_19_N_2_O_3_S [M + H]^+^ 367.11, found 366.95.

*N-(6-chloro-8H-indeno[1,2-d]thiazol-2-yl)-3,5-dimethoxybenzamide* (**7e**), eluting with DCM, yield = 39%; ^1^H NMR (400 MHz, DMSO-*d*_6_) δ 12.85 (s, 1H), 7.64 (d, *J* = 1.6 Hz, 1H), 7.54 (d, *J* = 8.0 Hz, 1H), 7.42 (dd, *J* = 8.0, 2.0 Hz, 1H), 7.33 (d, *J* = 2.0 Hz, 2H), 6.74 (t, *J* = 2.0 Hz, 1H), 3.94 (s, 2H), 3.84 (s, 6H) ppm. ^13^C NMR (100 MHz, DMSO-d6) δ 164.36, 162.60, 160.48, 154.14, 148.13, 135.82, 133.64, 131.60, 129.72, 126.87, 125.43, 118.83, 105.75, 105.13, 55.57, 32.49 ppm. MS (ESI): *m*/*z* calcd for C_19_H_16_ClN_2_O_3_S [M + H]^+^ 387.06, found 387.15.

*3,5-dimethoxy-N-(5-methoxy-8H-indeno[1,2-d]thiazol-2-yl)benzamide* (**4**), eluting with DCM, yield = 50%; ^1^H NMR (400 MHz, DMSO-*d*_6_) δ 12.80 (s, 1H), 7.44 (d, *J* = 8.4 Hz, 1H), 7.34 (d, *J* = 2.4 Hz, 2H), 7.07 (d, *J* = 2.4 Hz, 1H), 6.81 (dd, *J* = 8.4, 2.4 Hz, 1H), 6.73 (t, *J* = 2.4 Hz, 1H), 3.83 (s, 6H), 3.82 (s, 2H), 3.81 (s, 3H) ppm. ^13^C NMR (100 MHz, DMSO-*d*_6_) δ 164.32, 162.22, 160.50, 158.81, 138.17, 137.75, 133.77, 132.04, 125.68, 110.29, 105.74, 105.10, 104.65, 103.93, 55.58, 55.20, 31.64 ppm. MS (ESI): *m*/*z* calcd for C_20_H_19_N_2_O_4_S [M + H]^+^ 383.11, found 383.15.

*3,4,5-trimethoxy-N-(6-methoxy-8H-indeno[1,2-d]thiazol-2-yl)benzamide* (**7f**), eluting with DCM, yield = 44%; ^1^H NMR (400 MHz, DMSO-*d*_6_) δ 12.76 (s, 1H), 7.52 (s, 2H), 7.45 (d, *J* = 8.4 Hz, 1H), 7.21 (d, *J* = 2.4 Hz, 1H), 6.93 (dd, *J* = 8.4, 2.4 Hz, 1H), 3.89 (s, 6H), 3.86 (s, 2H), 3.80 (s, 3H), 3.75 (s, 3H) ppm. ^13^C NMR (100 MHz, DMSO-*d*_6_) δ 163.94, 162.29, 157.73, 154.98, 152.79, 147.95, 141.00, 130.07, 128.23, 126.73, 118.20, 112.35, 111.81, 105.61, 60.14, 56.11, 55.35, 32.41 ppm. MS (ESI): *m*/*z* calcd for C_21_H_21_N_2_O_5_S [M + H]^+^ 413.12, found 413.15.

*5-((6-methoxy-8H-indeno[1,2-d]thiazol-2-yl)carbamoyl)-1,3-phenylene diacetate* (**7g**), eluting with DCM, yield = 25%; ^1^H NMR (400 MHz, DMSO-*d*_6_) δ 12.91 (s, 1H), 7.86 (d, *J* = 2.0 Hz, 2H), 7.46 (d, *J* = 8.4 Hz, 1H), 7.33 (t, *J* = 2.0 Hz, 1H), 7.22 (d, *J* = 2.0 Hz, 1H), 6.94 (dd, *J* = 8.4, 2.0 Hz, 1H), 3.88 (s, 2H), 3.80 (s, 3H), 2.33 (s, 6H) ppm. ^13^C NMR (100 MHz, DMSO-*d*_6_) δ 172.06, 169.05, 164.01, 158.37, 157.74, 151.38, 147.94, 134.01, 130.00, 128.36,119.23, 118.29, 113.15, 112.35, 111.83, 55.35, 32.42, 20.86 ppm. MS (ESI): *m*/*z* calcd for C_22_H_19_N_2_O_6_S [M + H]^+^ 439.10, found 439.05

*3-methoxy-N-(6-methoxy-8H-indeno[1,2-d]thiazol-2-yl)benzamide* (**7h**), eluting with DCM, yield = 47%; ^1^H NMR (400 MHz, DMSO-*d*_6_) δ 12.81 (s, 1H), 7.72–7.70 (m, 2H), 7.48–7.44 (m, 2H), 7.22–7.18 (m, 2H), 6.94 (dd, *J* = 8.0, 2.4 Hz, 1H), 3.88 (s, 2H), 3.86 (s, 3H), 3.80 (s, 3H) ppm. ^13^C NMR (100 MHz, DMSO-*d*_6_) δ 164.43, 162.14, 159.31, 157.73, 155.00, 147.95, 133.25, 130.05, 129.78, 128.32, 120.45, 119.02, 118.26, 112.58, 112.34, 111.80, 55.41, 55.34, 32.40 ppm. MS (ESI): *m*/*z* calcd for C_19_H_17_N_2_O_3_S [M + H]^+^ 353.10, found 353.15.

*3-fluoro-N-(6-methoxy-8H-indeno[1,2-d]thiazol-2-yl)benzamide* (**7i**), eluting with DCM, yield = 34%; ^1^H NMR (400 MHz, DMSO-*d*_6_) δ 12.92 (s, 1H), 8.00–7.94 (m, 2H), 7.65–7.59 (m, 1H), 7.52 (dd, *J* =8.4, 2.4 Hz, 1H), 7.47 (d, *J* = 8.0 Hz, 1H), 7.22 (d, *J* = 2.4 Hz, 1H), 6.94 (dd, *J* = 8.0, 2.4 Hz, 1H), 3.88 (s, 2H), 3.80 (s, 3H) ppm. ^13^C NMR (100 MHz, DMSO-*d*_6_) δ 163.43,163.21, 161.90, 160.77, 157.77, 147.93, 134.26, 130.84 (d, *J* = 8.0 Hz), 129.95, 128.48, 124.36 (d, *J* = 3.0 Hz), 119.52 (d, *J* = 21.0 Hz), 118.30, 114.91 (d, *J* = 23.0 Hz), 112.36, 111.80, 55.34, 32.42 ppm. MS (ESI): *m*/*z* calcd for C_18_H_14_FN_2_O_2_S [M + H]^+^ 341.08, found 341.05.

*N-(6-methoxy-8H-indeno[1,2-d]thiazol-2-yl)thiophene-2-carboxamide* (**7j**), eluting with DCM, yield = 30%; ^1^H NMR (400 MHz, DMSO-*d*_6_) δ 12.92 (s, 1H), 8.28 (d, *J* = 8.0 Hz, 1H), 7.98 (d, *J* = 4.8 Hz, 1H), 7.45 (d, *J* = 8.0 Hz, 1H), 7.27 (t, *J* = 4.8 Hz, 1H), 7.22 (d, *J* = 2.4 Hz, 1H), 6.94 (dd, *J* = 8.0, 2.4 Hz, 1H), 3.87 (s, 2H), 3.80 (s, 3H) ppm. ^13^C NMR (100 MHz, DMSO-*d*_6_) δ 161.76, 159.35, 157.74, 154.97, 147.91, 137.30, 133.61, 130.69, 129.96, 128.64, 128.29, 118.24, 112.34, 111.78, 55.33, 32.43 ppm. MS (ESI): *m*/*z* calcd for C_16_H_13_N_2_O_2_S_2_ [M + H]^+^ 329.04, found 329.10.

*4-chloro-N-(6-methoxy-8H-indeno[1,2-d]thiazol-2-yl)benzamide* (**7k**), eluting with DCM, yield = 38%; ^1^H NMR (400 MHz, DMSO-*d*_6_) δ 12.90 (s, 1H), 8.14 (dt, *J* = 8.8, 2.0 Hz, 2H), 7.63 (dt, *J* = 8.4, 2.0 Hz, 2H), 7.47 (d, *J* = 8.4 Hz, 1H), 7.22 (d, *J* = 2.0 Hz, 1H), 6.94 (dd, *J* = 8.4, 2.4 Hz, 1H), 3.88 (s, 2H), 3.80 (s, 3H) ppm. ^13^C NMR (100 MHz, DMSO-*d*_6_) δ 163.77, 162.03, 157.74, 154.90, 147.93, 137.48, 130.82, 130.03, 129.97, 128.73, 128.38, 118.28, 112.34, 111.79, 55.34, 32.41 ppm. MS (ESI): *m*/*z* calcd for C_18_H_14_ClN_2_O_2_S [M + H]^+^ 357.05, found 356.90.

#### 3.1.2. Procedure for the Synthesis of Compound **7l**

To a solution of **5f** (528.2 mg, 3.0 mmol) in dry ethanol (10 mL) were added thiourea (456.7 mg, 6.0 mmol) and bromine (0.2 mL, 3.3 mmol) at room temperature. The reaction solution was stirred at 100 °C for 5–6 h. At the end of the reaction, the solvent was evaporated and aqueous ammonium hydroxide (25%) was added to the residue. The precipitated solid **6f** was collected without purification for the next step. The mixture of 6f (255.2 mg, 1.1 mmol), 3,5-dimethoxybenzoic acid (182.1 mg, 1.0 mmol), HATU (380.2 mg, 1.0 mmol), and DIPEA (0.5 mL 3.0 mmol) in DMF (6 mL) was stirred at room temperature for 2 h. The reaction mixture was quenched with water. The aqueous layer was extracted with EtOAc (20 mL × 2). The combined organic layers were dried over Na_2_SO_4_. The residue was purified by column chromatography on silica gel (eluting with DCM) to afford compound **7l** (103.0 mg, yield 26%) as a white solid.

^1^H NMR (400 MHz, DMSO-*d*_6_) δ 12.66 (s, 1H), 7.66 (dd, *J* = 8.4, 2.0 Hz, 1H), 7.32 (t, *J* = 2.0 Hz, 2H), 6.88 (s, 1H), 6.85 (dd, *J* = 8.4, 2.4 Hz, 1H), 6.73 (d, *J* = 2.4 Hz, 1H), 3.83 (s, 6H), 3.77 (s, 3H), 3.00–2.91 (m, 4H) ppm. ^13^C NMR (100 MHz, DMSO-*d*_6_) δ 164.31, 160.45, 158.42, 156.44, 143.66, 136.68, 133.90, 124.23, 123.33, 121.55, 114.08, 111.82, 105.75, 105.06, 55.59, 55.09, 28.65, 20.74 ppm. MS (ESI): *m*/*z* calcd for C_21_H_21_N_2_O_4_S [M + H]^+^ 397.12, found 396.95.

#### 3.1.3. General Procedure of Synthesis of **10a**–**10b** (Exemplified by **10a**)

A mixture of **8a** (10.0 mmol, 1.0 equiv), thiourea (20.0 mmol, 2.0 equiv), and iodine (10.0 mmol, 1.0 equiv) was stirred at 110 °C for 10 h. After the reaction was completed, the residue was triturated with MTBE and adjusted to pH 9–10 with 25% ammonium hydroxide. The precipitated solid was collected and washed with EtOAc (30 mL × 2) and NaHCO_3_ (15 mL × 2) aqueous solution. The separated organic layer dried over Na_2_SO_4_ and evaporated to dryness to afford crude product **9a**. The mixture of **9a** (3.3 mmol, 1.1 equiv), aromatic acid (3.0 mmol, 1.0 equiv), HATU (3.0 mmol, 1.0 equiv), and DIPEA (9.0 mmol, 3.0 equiv) in DMF (20 mL) was stirred at room temperature for 2 h. Then the reaction mixture was quenched with water. The aqueous layer was extracted with EtOAc (30 mL × 2). The combined organic layers were dried over Na_2_SO_4_. The residue was purified by column chromatography on silica gel (eluting with DCM) to afford compound **10a** as a white solid (406.7 mg, yield 35%). ^1^H NMR (400 MHz, DMSO-*d*_6_) δ 12.67 (s, 1H), 7.57 (d, *J* = 8.4 Hz, 1H), 7.32 (d, *J* = 2.0 Hz, 2H), 7.21 (s, 1H), 6.86 (d, *J* = 2.4 Hz, 1H), 6.83 (dd, *J* =8.4, 2.8 Hz, 1H), 6.74 (t, *J* = 2.4 Hz, 1H), 3.83 (s, 6H), 3.77 (s, 3H), 2.43 (s, 3H) ppm. ^13^C NMR (100 MHz, DMSO-*d*_6_) δ 164.74, 160.45, 158.60, 157.76, 149.01, 136.97, 134.22, 130.74, 127.31, 115.96, 111.22, 110.14, 105.80, 104.90, 55.58, 55.04, 21.26 ppm. MS (ESI): *m*/*z* calcd for C_20_H_21_N_2_O_4_S [M + H]^+^ 385.12, found 385.20.

*3,5-dimethoxy-N-(4-(4-methoxy-3-methylphenyl)thiazol-2-yl)benzamide* (**10b**), eluting with DCM, yield = 40%; ^1^H NMR (400 MHz, DMSO-*d*_6_) δ 12.70 (s, 1H), 7.77 (d, *J* = 2.4 Hz, 1H), 7.75 (s, 1H), 7.49 (s, 1H), 7.33 (d, *J* = 2.4 Hz, 2H), 6.99 (d, *J* = 8.8 Hz, 1H), 6.74 (t, *J* = 2.4 Hz, 1H), 3.84 (s, 6H), 3.82 (s, 3H), 2.20 (s, 3H) ppm. ^13^C NMR (100 MHz, DMSO-*d*_6_) δ 164.59, 160.46, 158.26, 157.15, 149.27, 133.86, 128.07, 126.72, 125.68, 124.62, 110.36, 106.38, 105.79, 105.09, 55.59, 55.31, 16.21 ppm. MS (ESI): *m*/*z* calcd for C_20_H_21_N_2_O_4_S [M + H]^+^ 385.12, found 385.25.

### 3.2. Molecule Docking

The protease structure, SARS-CoV-2 3CL^pro^ enzyme (PDB code: 6M2N) with 2.2 Å, was obtained from the the Protein Data Bank at the RCSB site (http://www.rcsb.org (accessed on 6 March 2022)). The molecule docking used the Lamarckian genetic algorithm local search method and the semiempirical free energy calculation method in the AutoDock 4.2 program. Additionally, the charge was added by Kollman in AutoDock 4.2, The docking methold was employed on rigid receptor conformation, all the rotatable torsional bonds of compound **7a** were set free, the size of grid box was set at to 10.4 nm × 12.6 nm × 11.0 nm points with a 0.0375 nm spacing and grid center (−33.798 −46.566 39.065), and the other parameters were maintained at their default settings.

### 3.3. Enzymatic Activity and Inhibition Assays

The enzyme activity and inhibition assays of SARS-CoV-2 3CL^pro^ have been described previously [20,36]. Briefly, the recombinant SARS-CoV-2 3CL^pro^ (40 nM at a final concentration) was mixed with each compound in 50 μL of assay buffer (20 mM Tris, pH 7.3, 150 mM NaCl, 1% Glycerol, 0.01% Tween-20) and incubated for 10 min. The reaction was initiated by adding the fluorogenic substrate MCA-AVLQSGFRK (DNP) K (GL Biochem, Shanghai, China), with a final concentration of 40 μM. After that, the fluorescence signal at 320 nm (excitation)/405 nm (emission) was immediately measured by continuous 10 points for 5 min with an EnVision multimode plate reader (Perkin Elmer, Waltham, MA, USA). The initial velocity was measured when the protease reaction was proceeding in a linear fashion; plots of fluorescence units versus time were fitted with linear regression to determine initial velocity. Plots of initial velocity versus inhibitor concentration were fitted using a four-parameter concentration–response model in GraphPad Prism 8 to calculate the IC_50_ values. All data are shown as mean ± SD, *n* = 3 biological replicates.

## 4. Conclusions

In summary, we synthesized a series of 8*H*-Indeno[1,2-*d*]thiazole derivatives and evaluated their biochemical activities against SARS-CoV-2 3CL^pro^. Among them, the representative compound **7a** displayed inhibitory activity with an IC_50_ of 1.28 ± 0.17 μM against SARS-CoV-2 3CL^pro^. Molecular docking elucidated that **7a** was well-docked into the binding pockets S1 and S2 of 3CL^pro^. These preliminary results could provide a possible opportunity for the development of novel inhibitors against SARS-CoV-2 3CL^pro^ with optimal potency and improved pharmacological properties.

## Data Availability

Data are available upon request to the Corresponding Authors.

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
