# Peer review of "Synthesis and Biochemical Evaluation of 8H-Indeno[1,2-d]thiazole Derivatives as Novel SARS-CoV-2 3CL Protease Inhibitors"

_molecules, 2022, doi:10.3390/molecules27103359_

Round 1

Reviewer 1 Report

This work is focused on the synthesis and characterization of a small collection of 8H-Indeno[1,2-d]thiazole derivatives and their in vitro evaluation as potential SARS-CoV-2 3CLpro inhibitors.

The results of the manuscript are structured in three parts. Related to the chemistry, the strategy followed for the design of the new analogs, the in vitro biological evaluation as potential enzyme inhibitors, and the prediction mode binding for the selected compound.

The manuscript, in my eyes, is interesting and attractive, especially in the development of new class of compounds with potential beneficial effect in the fight against SARS-CoV-2, and I encourage authors to continue this research

However, I have suggestions of things that should be addressed before its publication in Molecules. They are listed below:

1) The first one concerns to the numbers of the compounds. In the introduction, those previously described SARS-CoV-2 inhibitors are indicated with the name of the compound, excepting 13b, 2a and 8 that appear with these numbers. These are the numbers of the compounds in the original articles (indicated in the text with the appropriate bibliographic reference), but in this manuscript they should be indicated with a number of their own for this text. And they should start with number 1. This change will affect all the compounds, one that was employed as prototype (1 in the text) and all the rest. It is very difficult to read and to follow the text without a sequential order. Please check this and of course do the same in the Supplementary Information, as to the numbers of the compounds will change.

2) Lines 42-45. It is difficult to follow the sentence. The redaction of the sentence should be improved

3) Line 70. The sentence “from the appropriate ketones, thiourea” should be “from the appropriate ketones and thiourea

4) In Table 1 the structure of PF-07321332 (positive control) should appear, in the same way that the synthesized compounds described in the article, and the % inhibition of this compound. The IC50 of this compound is indicated in Table 1 but not the % inhibition determined by the methodology used by the authors. The positive control should be evaluated with the same methodology than the new compounds.

5) It is not indicated how many experiments were done to obtain the mean ± SD of the % inhibition and the IC50 values.

Related to this subject the description of the biological assays in the experimental section should be more extensive and there is no mention about the dose-dependent inhibition curves that authors should have developed to determine the IC50 values of the compounds.

6) In the experimental section, it is not indicated the eluent for the purification of the compounds. It should appear in the description of each compound. Also at this point, in the characterization of each compound, after the signal of each carbon there should appear a comma, without any preceding space, and followed by a space.

As example:

13C NMR (100 MHz, DMSO-d6) δ 164.25 , 162.15 , 160.54 , 157.20 , 155.04 , 147.96 , 133.85…

Should be: 13C NMR (100 MHz, DMSO-d6) δ 164.25, 162.15, 160.54, 157.20, 155.04, 147.96, 133.85…

The mass spectrometry data should be expressed with two decimals (as shown in the Supplementary material)

The comments of this point affect all the described compounds.

7) Concerning the virtual study of the interaction with the target, based on the fact that authors have found two compounds with high enzymatic inhibition (>70%) with low IC50, authors could have studied the interactions of both with the enzyme, not only compound 4b

8) Related to the potential antiviral activity, authors have not developed antiviral inhibition assays. It is known that inhibitors that exhibited inhibitory activities in biochemical protease assays, failed in cell-based viral infection assays. This study would improve the real potential of their compounds. Could authors perform this assay or at least state whether they plan to do it? I understand that it needs to have scientifical collaborations in case they cannot do it.

9)  Finally, in the table of contents of the Supplementary Material, for each compound it should be written: Spectral data for compound…

As it is explained it sounds reiterative.

In general terms I recommend this article for publication in Molecules with the suggestions described above, to be attended before its publication (major revisions)

Reviewer 2 Report

The manuscript “Synthesis and Biochemical Evaluation of 8H-Indeno[1,2-d]thiazole Derivatives As Novel SARS-CoV-2 3CL Protease Inhibitors” describes the synthesis of 8H-Indeno[1,2-d]thiazole derivatives and evaluation of biological activities against SARS-CoV-2 3CLpro. The manuscript deserves publication after major revision.

Suggestions for the authors to consider are listed below.

Abstract: Abstract is very concise.

Suggestion: Abstract may inform the reader about the scientific content of the manuscript.

Abstract: 3-chymotrypsin-like cysteine protease …

Suggestion: 3-Chymotrypsin-like cysteine protease …

Capital C for Chymotrypsin

Abstract:  ... 8H-Indeno[1,2-d]thiazole ...

Suggestion: … 8H-indeno[1,2-d]thiazole …

Lower letter for indeno

keywords:  8H-Indeno[1,2-d]thiazole derivatives; SARS-CoV-2; 3CLpro inhibitors

Suggestion: These keywords are present in the title of the manuscript. Keywords may be different from the words present in the title of the manuscript.

Line 30: … coronavirus disease 19 …

Suggestion: … coronavirus disease …

Line 43: … has been approved to combat to SARS-CoV-2 …

Suggestion: … has been approved to combat SARS-CoV-2 …

Line 44: … has attracted much attention form medicinal chemistries and pharmaceutical industry …

Suggestion: … has attracted much attention from medicinal chemists and pharmaceutical industry …

Line 69: … shown in Scheme 1, Scheme 2 …

Suggestion: … shown in Scheme 1 and Scheme 2 …

Line 70: … the appropriate ketones, thiourea …

Suggestion: … the appropriate ketone and thiourea …

Line 71-72: Ajusting methoxy group of compound 1 from 5 position to 6 position, we obtained compound 4b.

Suggestion: Adjusting the methoxy group of compound 1 from position 5 to position 6, afforded compound 4b.

Line 73-74: … replacement of the methoxy group with butoxy group, isobutoxy group, methyl group, chloro.

Suggestion: … substitution of the methoxy group for the butoxy, isobutoxy and methyl groups and for the chlorine atom.

Line 74-77: After replacing the 3,5-dimethoxybenzamido moiety on compound 4b with 3,4,5-trimethoxybenzamido moiety, 3,5-diacetoxybenzamido moiety, 3-methoxybenzamido moiety, 3-fluorobenzamido moiety, thiophene-2-carboxamido moiety and 4-chlorobenzamido moiety, compounds 4g - 4l were obtained.

Suggestion: After replacing the 3,5-dimethoxybenzamido moiety in compound 4b with 3,4,5-trimethoxybenzamido, 3,5-diacetoxybenzamido, 3-methoxybenzamido, 3-fluorobenzamido, thiophene-2-carboxamido and 4-chlorobenzamido, compounds 4g-4l were obtained.

Line 78: To evaluate the effect of ring enlargement and ring cleavage, compounds 4m, 7a and 7b were synthesized.

Suggestion: To evaluate the effect of ring enlargement and ring formation compounds 4m, 7a and 7b were synthesized.

Comment: There was no ring cleavage in the production of 4m and 7a.

Line 93-96: By replacing methoxy group on compound 4b with butoxy group (4c), isobutoxy group (4d), methyl group (4e), chloro (4f), the inhibitory activities of corresponding compounds 4c 4f significantly dropped compared with compound 4b.

Suggestion: By replacing methoxy group on compound 4b with butoxy (4c), isobutoxy (4d), and methyl groups (4e) and chlorine atom (4f), the inhibitory activities of the corresponding compounds 4c4f significantly dropped compared to compound 4b.

Line : These results demonstrated that the effects of steric hindrance and electron withdrawing were clearly detrimental to inhibitory activities.

Suggestion: These results demonstrated that the effects of steric hindrance were clearly detrimental to inhibitory activities.

Comment: Comparing methoxy to butoxy, isobutoxy, and methyl groups, no increase in electron withdrawing is observed. Only steric hindrance. Comparing methoxy group to chlorine atom no electron withdrawing effect is observed.

Line 124: surf representation of the compound 4b …

Suggestion: What is surf representation?

Line 134: General procedure for synthesis of compounds 4b - 4m (exemplified by 4b)

Suggestion: General procedure for the synthesis of compounds 4b - 4m (exemplified by 4b)

Line 135: A solution of 2b (6.2 mmol, 1.0 equiv) in dry ethanol (25 mL) were added thiourea …

Suggestion: To a solution of 2b (6.2 mmol, 1.0 equiv) in dry ethanol (25 mL) were added thiourea …

Line 137: After the reaction completed, the reaction mixture was evaporated and then neutralized with 25% ammonium hydroxide.

Suggestion: At the end of the reaction, the solvent was evaporated and aqueous ammonium hydroxide (25%) was added to the residue.

Line 151: Yield = 32.1%;

Suggestion: Yield = 32%;

Comment: Quote reaction yield all over the main text without decimal places.

Line 151: 5a (10.0mmol, 1.0 equiv), thiourea (20.0 mmol, 2.0 equiv), iodine (10.0mmol, 1.0 229 equiv) were stirred at 110℃ for 10h.

Suggestion: A mixture of 5a (10.0mmol, 1.0 equiv), thiourea (20.0 mmol, 2.0 equiv), and iodine (10.0mmol, 1.0 229 equiv) was stirred at 110℃ for 10h.

Reviewer 3 Report

This manuscript describes the synthesis of indeno thiazole type compounds & SAR discussion for SARS-CoV-2 3CL protease inhibitors.  Several analogues (14 examples) were synthesized in 2 steps utilizing conventional Hantzsch-type thiazole synthesis.  Judging from the SAR data in Table 1, the scope of the compound such as substituent effect and ring size looks like to be extremely too narrow; Only two compounds exhibited significant activity [IC50 (μM) = ca. 1~3].  The first lead compound 4b found by the HTS screening was best and the others were almost inactive. 

The reviewer considers that the motivation for this project and the reason for selection of indeno thiazole type compound should be more clearly described. 

Characterization of these compounds were well-performed.  Additional integration data of the 1H NMR will be more appropriate.     

On the whole, the reviewer recommends the publication in Molecules after minor revisions commented below.

<Comments and suggestions>

  1. The reader wants to know the level of bioactivity compared with the representative drugs as shown in Figure 1. Please indicate this issue.
  2. Is the HTS screening was implemented through your original library? Please refer the original sores. 
  3. Synthetic method of six-membered ring should be provided.
  4. Insert one space between the number and the unit throughout the text. Carefully check this issue.   ###°C, ##h → ### °C, ## h., etc.

Round 2

Reviewer 1 Report

All  suggestions have been attended.

I recommend this article for publication 

Reviewer 2 Report

The manuscript “Synthesis and Biochemical Evaluation of 8H-Indeno[1,2-d]thiazole Derivatives As Novel SARS-CoV-2 3CL Protease Inhibitors” describes the synthesis of 8H-Indeno[1,2-d]thiazole derivatives and evaluation of biological activities against SARS-CoV-2 3CLpro. The manuscript deserves publication after extensive editing of English language.

Abstract: Abstract is very concise.

Suggestion: Abstract may inform the reader about the scientific content of the manuscript.

keywords:  8H-Indeno[1,2-d]thiazole derivatives; 3CLpro inhibitors

Suggestion: These keywords are present in the title of the manuscript.

Suggestion: 8H-Indeno[1,2-d]thiazole derivatives can be replaced by COVID-19; 3CLpro inhibitors can be replaced by Mpro inhibitors.
